# Lattice Boltzmann-Discrete Element Modeling Simulation of SCC Flowing Process for Rock-Filled Concrete

**DOI:** 10.3390/ma12193128

**Published:** 2019-09-25

**Authors:** Song-Gui Chen, Chuan-Hu Zhang, Feng Jin, Peng Cao, Qi-Cheng Sun, Chang-Jun Zhou

**Affiliations:** 1Tianjin Research Institute for Water Transport Engineering, Ministry of Transport of the People’s Republic of China, Tianjin 300456, China; chensg05@163.com; 2China Gezhouba Group Three Gorges Construction Engineering Co., LTD, Yi Chang 443002, China; zch07328@163.com (C.-H.Z.); qcsun@tsinghua.edu.cn (Q.-C.S.); 3Department of Hydraulic Engineering, State Key Laboratory for Hydroscience and Engineering, Tsinghua University, Beijing 100084, China; 4College of Architecture and Civil Engineering, Beijing University of Technology, Beijing 100124, China; caopeng518888@126.com; 5School of Transportation and Logistics, Dalian University of Technology, Dalian 116023, China; zhouchangjun@dlut.edu.cn

**Keywords:** rock-filled concrete, self-compacting concrete, lattice Boltzmann method, discrete element method, passing factor

## Abstract

Since invented in 2003, rock-filled concrete (RFC) has gained much attention and has been successfully applied in more and more civil and hydraulic projects in China. This study developed a numerical framework to simulate self-compacting concrete (SCC) flows in the voids among rocks of RFC, which couples the lattice Boltzmann method and discrete element method (DEM). The multiple-relaxation-time scheme is used to simulate self-compacting mortar (SCM) for better stability while the motion of coarse aggregates in SCC is simulated with DEM. The immersed moving boundary method is incorporated to deal with the interactions between coarse aggregates and SCM. After validation, the coupled framework is applied to study SCC flows in a single channel and in porous media with multi-channels. A passing factor *PF* was proposed and calculated to describe quantitatively the passing ability of SCC through a single channel. The study found that jamming of SCC occurs when the ratio *Ar* of the gap width to particle diameter is smaller than 2.0 and the jamming risk increases with solid particles fraction while the passing ability has a weak relation with the pressure gradient. Further, SCC flow is self-tuning in porous media with multi-channels and it is prone to go through larger or wider gaps. Exceeded existence of narrow gaps will significantly increase the jamming risk.

## 1. Introduction

Rock-filled concrete (RFC), invented by Jin and An [1], is an innovative construction technology of mass concrete. By the end of 2017, the total amount of RFC used in engineering has exceeded 2,000,000 m^3^ with a broader and broader application prospect.

RFC mainly consists of two parts: rock skeleton (called rockfill) and self-compacting concrete (SCC) [2]. SCC with good fluidity can fill the voids among rocks well driven by gravity, resulting in a dense concrete after SCC hardens. However, jamming may occur when SCC flows through structures with small space [3,4]. The improper filling or loose structure will be harmful for mechanical properties and durability of RFC. Xie et al. [5] conducted an experiment study of fluidity of SCC on the properties of RFC. Wang et al. [6] studied the effects of casting procedures on the properties of RFC, such as compressive strength, water permeability, and porosity of the interfacial transition zone (ITZ) between SCC and rocks. However, detailed information in the filling process has not been revealed due to the observation difficulty in physical experiments. Numerical modeling can not only reproduce the filling process of SCC in rock skeletons, but also obtain the whole velocity field and other information as well as the interaction mechanism of jamming, which is rather important and useful for further study of RFC and SCC. 

The integration and mechanical properties of RFC are greatly related to the fill effect of SCC. During casting the RFC, when the air voids among rocks are not large enough, fine sand in SCC may be stuck in between, which would weaken the fill effect of SCC. Therefore, efforts are urged to study the flow characteristics and jamming mechanism of SCC.

Enormous approaches have been developed to study the fluidity of SCC. They can also be divided into three categories: the single fluid model, the discrete particle model, and the fluid-particle model. At the macroscopic scale, SCC can be regarded as a continuous Bingham fluid or other non-Newtonian fluid. Based on this simplification, 3D simulations of different slump test and fresh concrete flow around reinforcing bars were studied by some researchers [3,4,7,8]. However, the single fluid model cannot be used to simulate the process of jamming and aggregate segregation. At the micro scale, SCC can be discretized into a large number of spherical particles, thus the discrete element model (DEM) was used to simulate SCC flowing through rebars [9,10] or predict concrete rheological properties [11,12]. The disadvantages of DEM are that the physical foundation is not clear and contact parameters among particles depend on physical tests. Combining the characteristics of the aforementioned methods, the fluid-particle two phase model regards SCC as a homogeneous fluid (self-compacting mortar) containing suspended particles (coarse aggregates). Moresi et al. [13] proposed a finite element method with Lagrangian integration points (FEMLIP) for large deformation modeling of viscoelastic geomaterials. Dufour and Pijaudier-Cabot [14] used FEMLIP to simulate concrete flow inside Lbox. The most commonly used methods for liquid phase of SCC are traditional computational fluid dynamic (CFD) methods like finite element method (FEM) and finite volume method (FVM). However, they are not suitable for RFC since complex stationary boundaries (voids in rockfill) and moving boundary (coarse aggregates) are involved. The lattice Boltzmann method (LBM) can deal with complex boundary conditions simply and straightly [15,16,17]. As a new computational fluid dynamics method, LBM has great advantages over traditional numerical methods. For example, the evolution equation is simple, stable and easily parallel computing. As to interactions between coarse aggregates, it can be substituted with particle–particle interaction, which can be solved with the mature and professional discrete element method (DEM). Thus, the combination of LBM and DEM is an ideal tool for numerical study of simulation of the SCC flowing process for RFC. 

From the introduction on the simulation approaches on non-Newtonian fluid above, it is found that the LBM has advantages in simulating the flow behavior of SCM. The LBM adopts explicit methods to calculate the flow behavior of fluid. The LBM does not require too dense element meshing. And the robustness of the convergence in LBM is better. Therefore, the LBM is capable to describe the flow of SCM in rock skeleton. Numerous studies were conducted and satisfied results were achieved [18,19]. However, the LBM cannot analyze the jamming phenomenon of fine aggregate. On the other hand, the DEM is capable to describe particle movements. Wang et al. [20], Wu et al. [21], Cao et al. [22], and Wu et al. [23] characterized the different shapes of aggregates with DEM. Zhang et al. [24] and Owen et al. [25] studied the collision and movement of polygonal aggregates with DEM. 

Based on the analyses above, this study developed a numerical framework to simulate SCC flowing in RFC, which couples LBM and DEM. The multiple-relaxation-time (MRT) scheme is adopted to simulate self-compacting mortar (SCM) for better stability while the motion of coarse aggregates is simulated with DEM. The immersed moving boundary method is included to deal with interactions between fluid flow and moving particles. The numerical method is briefly introduced in Section 2 while validation of the framework is presented in Section 3. Then the coupled framework is further applied to study SCC flows in a single channel and in porous media with multi-channels, results of which are shown in Section 4 while Section 5 concludes and give some outlooks for further study.

## 2. Coupled LBM-DEM

The multiple-relaxation-time lattice Boltzmann model (MRT-LBM) [15,16] was applied in our work since it overcomes some obvious defects of the Lattice Bhatnagar-GGross-Krook (LBGK) model [26,27,28,29]. Based on the continuous Boltzmann equation, the generalized lattice Boltzmann equation for MRT-LBM can be expressed as Equation (1):(1)$$f(x+c_{i}\Delta t,t+\Delta t)=f(x,t)-M^{-1}S(m-m^{eq})$$
where, **c***_i_* represents the velocity of a particle, **f**(**x**,*t*) represents the density distribution function of a particle moving with a speed of **c***_i_* at position *x* and at the moment of *t*; **M** represents the transformation matrix transforming **f** and **f***^eq^* into the moment space, with **m**
*=*
**M***·***f**. **m** and **m***^eq^* represent the moment vector and equilibrium moment vector, respectively. The diagonal matrix $S=diag(0,s_{e},s_{\varepsilon },0,s_{q},0,s_{q},s_{v},s_{v})$ is the relaxation time for corresponding moment vectors. In this study, to reach an optimized stability of the model, *s_e_* = 1.1, *s_ε_* = 1.0, *s_q_* = 1.2 while *s_v_* can be obtained by the lattice kinematic viscosity *υ*, see Equation (2):(2)$$\nu =\frac{1}{3}\left(\frac{1}{s_{\upsilon }}-\frac{1}{2}\right)$$

Other details can be found in our previous works [27,28].

Self-compacting concrete can be regarded as a two-phase Bingham material, in which self-compacting mortar is the Bingham fluid phase while coarse aggregates are the solid phase. To obtain more stable results, the behavior of SCM is well described by Papanastasiou modified Bingham model [30], which is expressed as Equation (3):(3)$$\tau =\left(\frac{\tau_{0}}{\left|\dot{\gamma }\right|}\left[1-e^{-m\left|\dot{\gamma }\right|}\right]+\mu_{p}\right)\dot{\gamma }$$
where, *τ* is the shear stress, $\dot{\gamma }$ is the shear rate, *τ*_0_ is the yield stress, and *μ_p_* is the plastic viscosity. The apparent viscosity of Bingham fluids can be deduced from Equation (4):(4)$$\mu =\frac{\tau }{\dot{\gamma }}=\mu_{p}+\frac{\tau_{0}}{\left|\dot{\gamma }\right|}\left[1-e^{-m\left|\dot{\gamma }\right|}\right]$$
where $\left|\dot{\gamma }\right|=\sqrt{2\Pi_{\dot{\gamma }}}={\left[2\sum_{\alpha ,\beta =1}^{l}S_{\alpha \beta }S_{\alpha \beta }\right]}^{1/2}$ and $\Pi_{\dot{\gamma }}$ is the second invariant of the strain rate tensor. *S_αβ_* can be locally gained from the non-equilibrium part of the function $f_{i}^{\left(1\right)}(x,t)=f_{i}(x,t)-f_{i}^{eq}(x,t)$, that is Equation (5):(5)$$S_{\alpha \beta }=-\frac{1}{2\rho c_{s}^{2}\delta t}\sum_{i=0}^{8}c_{i\alpha }c_{i\beta }\sum_{j=0}^{8}{\left(M^{-1}SM\right)}_{ij}f_{i}^{\left(1\right)}(x,t)$$

Therefore, the apparent viscosity *μ* can also be locally calculated, and based on Equation (2), the relaxation factor *s_v_* can be derived. 

As to DEM, the soft contact model is applied to account for the jamming effect of SCC aggregates, which is composed of the normal force and tangential force. For particle *i* and particle *j*, which are located at *X_i_*, *X_j_* with velocities *V_i_*, *V_j_*, respectively, the contact forces can be calculated as Equation (6):(6)$$\begin{array}{l} F_{n}=K_{n}\delta_{n}^{}+\gamma_{n}\frac{d\delta_{n}^{}}{dt}\\ F_{t}=K_{t}\delta_{t}^{}+\gamma_{t}\frac{d\delta_{t}^{}}{dt} \end{array}$$
where, *F* is contact force, *K* is stiffness, *δ* is the normal overlap or tangential displacement, and *γ* is viscous coefficient. The subscript *n* and *t* represent normal and tangential directions, respectively. 

For the coupling strategy, the immersed moving boundary method proposed by Noble and Torczynski [31] is applied, which developed a more accurate and smooth lattice representation of particles to reduce the fluctuation of the computed hydrodynamic forces [32]. The hydrodynamic force exerted on a solid particle by liquid can be gained by adding the momentum change of all the boundary nodes together, Equation (7):(7)$$F_{f}=\sum_{1}^{n}\left(B_{s}\sum_{1}^{Q}\Omega_{i}^{s}c_{i}\right)$$
where, *n* represents the total number of boundary nodes and *Q* = 9 for D2Q9 model. The additional collision term, Ω*_i_^s^*, is introduced to account for the effect of solid on the fluid. *B_s_* is the weighting function for the additional collision operator Ω*_i_^s^* based on *ε_s_*, which is the area fraction of the nodal cell covered by a solid particle. 

***F****_f_* is further applied in DEM to determine a particle’s position and velocity in Equation (8):(8)$$ma=F_{c}+F_{f}+mg$$
where, *m* is the mass of particle, subscript *c* is the contact force between particles. The subscript *f* is the force of fluid loaded on particles. The interactions between particles and fluid are bidirectional, which is important for our problem.

## 3. Validation on Numerical Model

The circular particles’ sedimentation due to gravity in a viscous fluid is studied to verify the LBM-DEM coupling framework.

### 3.1. Two Circular Particles’ Sedimentation in Newtonian Fluid

To compare the numerical results with numerical case from benchmark literature [33,34] combining desired elements of the immersed boundary method, the direct forcing method and the lattice Boltzmann method for the simulation of particulate flows, parameters and boundary conditions are set below. The channel is set as 0.02 m wide and 0.08 m high. The acceleration of gravity, *g*, equals to 9.8 m/s^2^. The fluid is assumed as water, i.e., 1.0 × 10^−6^ m^2^/s viscosity and 1000 kg/m^3^ density. The particles’ radius is 0.001 m and they own a density of 1010 kg/m^3^. The lattice spacing Δ*x* is set as 10^−4^ m, so the computational domain is 200 × 800 in lattice units. Δ*t_LBM_* = 5 × 10^−4^ s while Δ*t_DEM_* = 2.5 × 10^−4^ s. Thus, according to Equation (2), the viscosity in lattice unit *υ* is 0.05 and the relaxation factors *s_v_* = 1.538. The contact stiffness *K* equals to 10^−4^ kg/s^2^. The initial locations of the two particles are (99.9, 720) and (100, 680) in lattice units, respectively. Periodic boundaries are assumed for the top wall and the bottom wall while the bounce-back condition is assumed for the two side walls. Both particles and fluid initially have no velocity. The two particles begin settling in the y-direction driven by the gravity.

Figure 1 shows the positions of two particles as well as the flow field at seven different times. It must be pointed out that the Drafting, Kissing and Tumbling (DKT) motion that is observed in Figure 1 was also observed by Patankar et al [35]. As illustrated in Figure 1, in the beginning, because of the influence of the wake of the bottom particle, the resistance due to water to the upper particle is smaller, causing a larger speed than the bottom particle. Thus, the upper particle will slowly catch up with the bottom one. Since the two particles are initially slightly offset in the *x* direction, when the contact occurs, tumbling occurs and the upper particle with higher speed will gradually go over the bottom particle. This phenomenon is the typical drafting, kissing, and tumbling (abbreviated as DKT) process from Bakhshian and Sahimi [36], and Bakhshian et al. [37]. 

The time histories of position and velocity of the two particles are shown in Figure 2 and Figure 3. The drafting process is before 1.0 s, and kissing process is from 1.0 s to 2.5 s, and tumbling process is from 2.5 s to 3.5 s. The results from the present method and Feng et al. [33] match very well before kissing while after tumbling some deviation shows up. The DKT process is a nonlinear instability problem and the tumbling process is very sensitive to numerical methods adopted, causing errors between different algorithms [35]. The reproduction of the DKT process and the comparison between the results demonstrate the validation of the presented coupled LBM-DEM procedure in Newtonian fluid.

### 3.2. Circular Particles’ Sedimentation in Bingham Fluid

The sedimentation process of particles in Bingham fluid is further simulated. For Bingham fluids, results are often expressed as the Bingham number (*Bn*), defined as Equation (9):(9)$$Bn=\frac{\tau_{0}L_{c}}{\mu_{p}U_{c}}$$
where, *L_c_* and *U_c_* represent the characteristic length and velocity, respectively. Note that circular particles are investigated, the characteristic length is the diameter *d*. The characteristic velocity is the Stokes’ velocity of the particle settling in a fluid with viscosity equal to Bingham fluid. The Stokes’ velocity is expressed as Equation (10):(10)$$u_{stokes}=\frac{(\rho_{p}-\rho_{f})gd^{2}}{18\mu }$$
where, $\rho_{f}$ fand $\rho_{p}$ represent densities of fluid and particle, respectively. *μ* describes the fluid’s dynamic viscosity, equal to plastic viscosity *μ_p_* of the Bingham fluid.

All results are transformed into dimensionless for the comparison purpose. Velocity is normalized by the Stokes velocity while time is divided by *d/u_stokes_*, which represents the time cost to travel a *d* distance with *u_stokes_*.

Firstly, the sedimentation of one particle was simulated to validate the coupled LBM-DEM approach for the Bingham fluid. The parameters are the same with Section 3.1, as well as boundary conditions. An exception is that the fluid is Bingham fluid. The yield stress $\tau_{0}$ is varied to gain different *Bn*s while other parameters remain constant.

From Figure 4, when *Bn* = 0, the fluid is equivalent to a Newtonian fluid, the circular particle will reach the Stokes’ velocity. This validates the coupled algorithm from another perspective. As *Bn* increases, the terminal velocity drops down, dragged by the escalating yield stress. It takes less time to reach the highest velocity as the *Bn* increases. In a Bingham fluid, a particle has to overcome not only the viscous force, but also a resistance due to the yield stress. Such resistance from yield stress consumes parts of the potential energy from the gravity. And the more potential energy consumed, the less time is required to reach the final velocity and smaller terminal speed can be achieved.

Then, sedimentation of two circular particles in the Bingham fluid are simulated, as shown in Figure 5, Figure 6 and Figure 7. The standard DKT process can be observed when *Bn* is small, similar in Figure 1, which validates the coupling of the LBM-DEMB in Bingham fluid. The snapshots at the kissing time of the sedimentation processes with different Bingham numbers are shown in Figure 5. The position where the two falling particles kiss is lower and the time needed increases if the *Bn* becomes larger. When the *Bn* is larger enough, the top particle (particle 1) will never catch up with the bottom one (particle 2). They fall at the same velocity and their distance remains constant, as shown in Figure 7. Different from Newtonian fluid, the DKT process disappears in Bingham fluid, as shown in Figure 5 and Figure 6. Additionally, the final speeds of the two particles decrease as *Bn* increases. The distances the two particles travel will be shortened.

These dissimilarities reflect the distinct nature of Newtonian and Bingham fluids. In Newtonian fluid, falling particles need solely face the resistance from viscosity, i.e., drag force. However, in Bingham fluids yield stress will be added as well. More resistance means more energy consumed and smaller velocity. As a result, the shear stress induced is not large enough to exceed the yield stress and the area of yielded region deceases with the *Bn* increasing. 

## 4. Simulation of SCC Flow Process

The validated framework was further applied to study SCC flows in a single channel and porous media with multi-channels and the influencing factors are investigated.

### 4.1. Study of SCC Flow in a Single Channel

The two-phase Bingham flow through a single channel is modeled with the coupled LBM-DEM framework and the impact factors of the passing ability of two-phase Bingham flow are studied. The simulation layout is shown in Figure 8. *L* = 80, *H* = 100 (all dimensions are in lattice units). For simplicity, circular particles are used to represent coarse aggregates in SCC, whose diameter *d_particle_* = 8. Pressure periodic boundary condition is applied in both the upper and bottom boundaries while halfway bounce-back boundary condition is used in the vertical walls. The larger particles are fixed whose diameters varied to create different gaps. In terms of parameters in Papanastasiou’s model, plastic viscosity *μ_p_* = 0.4, yield stress *τ_0_* = 2.0 × 10^−6^, the power index *m* = 1 × 10^8^. The passing ability of two-phase Bingham fluid-particle flow in a single channel is related to the pressure gradient, the volume fraction of solid particles and the width of the gap. For convenience, three dimensionless parameters are defined as follows Equations (11)–(13):(11)$$Gn=Gd/\tau_{0}$$
(12)$$\phi_{s}=V_{s}/(V_{s}+V_{f})$$
(13)$$Ar=d_{gap}/d_{particle}$$
where *G* is the pressure gradient imposed between the upper and bottom sides, *τ_0_* is the yield stress of Bingham fluid, and *V_s_, V_f_* are volume fractions of solid particles and fluid, respectively. *d_gap_* and *d_particle_* are the width of gap and the diameter of moving particles, respectively. 

Obviously, both the fluxes of solid and fluid through cross section *S* will decrease with diminishing passing ability. In order to describe the passing ability of two-phase flow through a single channel quantitatively, a parameter *PF* is proposed, which is defined as Equation (14):(14)$$PF=PF_{s}+PF_{f}=Q_{s}/\left(Q_{s0}+Q_{f0}\right)+Q_{f}/\left(Q_{s0}+Q_{f0}\right)=\left(Q_{s}+Q_{f}\right)/\left(Q_{s0}+Q_{f0}\right)$$
where, *PF_s_* and *PF_f_* are passing factors of solid particles and fluid, *Q_s0_* and *Q_f0_* are fluxes of particles and fluid through cross section *S* when there is no fixed particle, while *Q_s_* and *Q_f_* are fluxes of particles and fluid through cross section *S* with different *Ar*. The passing ability of two-phase flow is excellent and no jamming of moving particles occurs when *PF* = 1. However, the risk of jamming will increase with *PF* decreasing while jamming of particles will definitely emerge when *PF* ≈ 0. In order to eliminate the deviation, the fluxes of particles and fluid are calculated every 10^5^ time steps and the total time steps are 3 × 10^6^, the average value of the 30 results is used as the final average fluxes of particles and fluid. The simulation result when *Ar* = 8 is shown in Figure 9.

In total, 135 cases were simulated with *Ar = 1.3~10*, *ϕ_s_* = 0.46, 0.35, 0.25, and *Gn* = 67,133,266, respectively. The passing factor of the Bingham fluid-solid flow in every case is calculated and presented in Figure 10. It can be seen that *Ar* is the predominant parameter that has an important influence on the passing ability and the passing factor suffers a sharp decrease when Ar < 2 for all cases, which is in accordance with results in Martys [10]. When *Ar* is close to 1, the passing factor increases to a certain extent, because the arch structure is difficult to form when the gap is too narrow. Pressure gradient has little effect on the jamming status of Bingham fluid-solid flow.

The passing factors versus *Ar* with same pressure gradient *Gn* but different *ϕ_s_* are shown in Figure 11. *ϕ_s_* has an effect on the passing factor to a certain extent. The average passing factors when *Ar* < 2 for different *ϕ_s_* are presented in the subgraphs of Figure 11. It is clearer from the Figure 11 that the passing factor decrease with increasing *ϕ_s_*, which means it is more likely to jam with more particles. 

On the other hand, the *PF_f_* and *PF_s_* are also calculated for all the cases, which is showed in Figure 12. Both the fluxes of fluid and particles decreases with *Ar* while the flux of particles decreases more rapidly than that of fluid when *Ar* < 2. This indicates when the gap is relatively narrow and jamming is prone to occur, more fluid would flow through the gap than particles and fill the pores behind the gap. This is the same in SCC and RFC. When particles in SCC are blocked by the stationary rocks and cannot get through the gap, Bingham fluid or SCM can still get through and fill the pores behind the jamming gap. That is why the space between large rocks can be filled well in rock-filled concrete. 

Further, the solid volume fraction of the two-phase Bingham fluid-solid flow after the cross section *S* is calculated, which is defined as Equation (15):(15)$$\phi_{s}^{p}=Q_{s}/\left(Q_{s}+Q_{f}\right)=PF_{s}/PF$$

The results of $\phi_{s}^{p}$ are shown in Figure 13. For the two-phase Bingham fluid-solid flow through a single channel, there are three stages: free flow (*Ar* > 3.5), $\phi_{s}^{p}$ keeps constant and equals; slightly jamming (2 < *Ar* < 3.5), both $\phi_{s}^{p}$ and the passing ability of particles start to decrease because of the existence of the gap while the passing ability of fluid does not change much; jamming (*Ar* < 2), both the passing abilities of particles and fluid decrease sharply. Besides, amplitude of variation of the $\phi_{s}^{p}$ also decreases with the original solid fraction, which implies smaller solid fraction means smaller jamming risk of the two-phase flow. 

It can be concluded that gap and diameter ratio *Ar* plays the most important role in the passing ability of two-phase Bingham fluid-solid flow through a single channel, which is mainly because the blockage of the two-phase flow is caused by the block of particles in nature. For dense granular flow, To et al. [38] found that when the channel size is smaller than 3 to 4 times of the maximum particle diameter, it is easy to form particles, arch and jamming occurs. For two-phase flow problem, the presence of the liquid phase reduces the contact force between the particles, therefore, the gap and diameter ratio *Ar* for block or jamming becomes smaller. The increase of the solid particles fraction will raise the probability of forming arches of granular particles, thus increasing the blockage risk and decreasing the passing ability. Once the blockage occurs, the increase in the pressure gradient usually cannot break the particles arch, so the pressure gradient does not affect the passing performance of the two-phase flow.

Note that the presented results are similar to those of Roussel et al. [4], which brought out a probabilistic approach to evaluate the passing ability of fresh concrete through obstacles. It is found that the probability of granular jamming of SCC crossing a flow contraction increases with the number of particles crossing the obstacles, their volume fraction and the ratio between the diameter of the particles and the contraction gap.

### 4.2. Study on Bingham Two-Phase Flow in Porous Media

This section focuses on the two-phase Bingham flow in porous media. The solid fraction is set to *ϕ_s_* = 0.46, while *Gn* = 166. A number of simulations were conducted for flows in porous media containing various narrow channels (*Ar* < 2). The flow fields at different times are shown in Figure 14, Figure 15 and Figure 16.

The study found that two-phase Bingham flow in porous media is self-tuning. The flow is prone to go through relatively wider channels, leading to the existence of “main stream” (red channel in Figure 14) and “tributary stream” (blue channel in Figure 14) in porous media. The flow rate at “main stream” is larger and most of the two-phase flow go through the porous medium via these main channels while the flow rate at the “tributary stream” is much smaller, especially in the direction perpendicular to the pressure gradient, the velocity is almost zero. The regions around “tributary stream” in RFC is the place where blockage is most likely to occur and the filling effect is not good. In addition, with the increase in the number of narrow channels, both the average flow velocity and passing ability of the two-phase flow decrease and the decrease of velocity makes particles jamming more easily. 

## 5. Conclusions

This study presents a coupled LBM-DEM approach for modeling SCC flows in RFC, which incorporates the immersed moving boundary method for interactions between fluid and particles. Fluid is described with the more stable MRT-LBM while particles’ movement is solved by DEM. The proposed method is validated with the particles’ sedimentation in Newtonian and Bingham fluids.
The coupled LBM-DEM approach proposed is capable of simulating the flow of SCM in RFC with the particle collision in SCC considered. Compared with literature, the approach proposed exhibited accuracy. More importantly, it can cover the nonlinear instability of particle flow due to the collisions among particles in SCC.When the SCM was treated as the Bingham flow, according to the proposed coupled LBM-DEM approach, the coarse aggregates in SCC moved much slower and the viscous damping phenomenon was observed. These may be the mechanism behind the jamming among rocks in RFC.It is found that the key factor that controls jamming is the gap width to particle diameter ratio *Ar*. The critical *Ar* is about 2.0, below which the risk of jamming increases sharply.As the volume fraction of solid particles increases, jamming becomes easier.The pressure gradient has little impact on the jamming status.In the multi-channel case, SCC is able to ‘find’ the wider gaps automatically. As a result, the flow can be divided into the “main channel” and “tributary stream”. The risk of jamming increases significantly as the number of narrow gaps increases.

To summarize, the coupled LBM-DEM is a powerful and efficient numerical method to study the mesoscopic mechanisms of SCC’s casting process. The flow of SCC in rockfill of RFC is three dimensional in nature. A 3D numerical study of SCC jamming in RFC will be presented in the future. 

## Figures and Tables

**Figure 1 materials-12-03128-f001:**
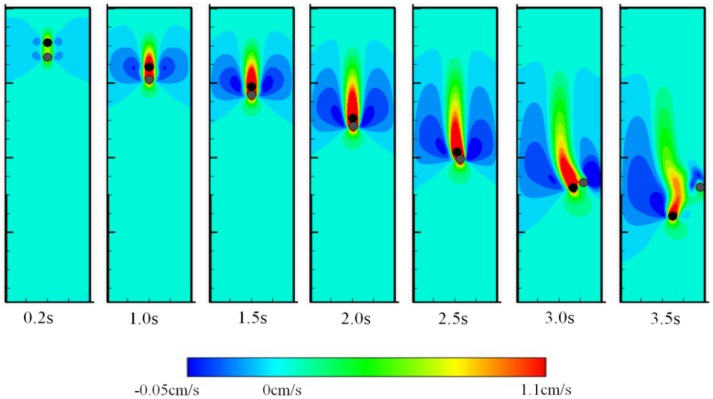
The sedimentation process of two circular particles in a channel and corresponding flow fields at different time stages.

**Figure 2 materials-12-03128-f002:**
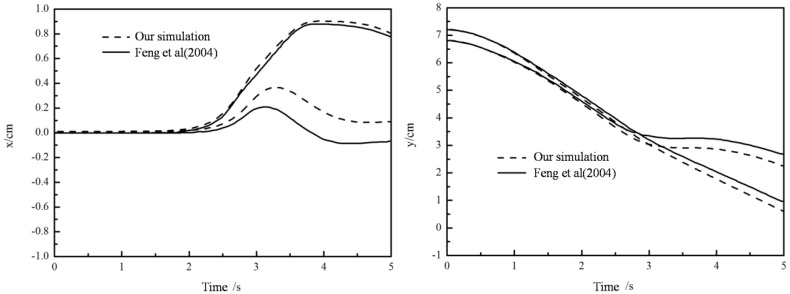
Coordinates of the two particles (left: *x* direction; right: *y* direction).

**Figure 3 materials-12-03128-f003:**
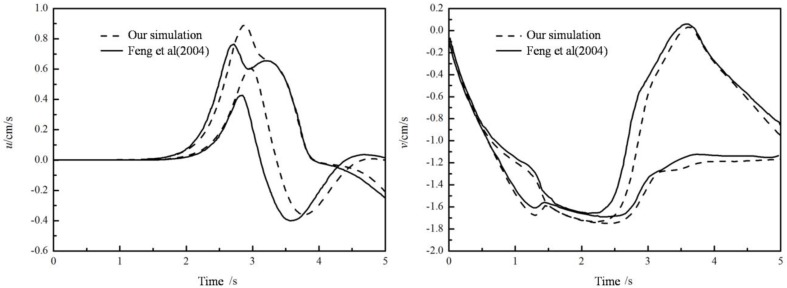
Velocities of the two particles (left: *x* direction; right: *y* direction).

**Figure 4 materials-12-03128-f004:**
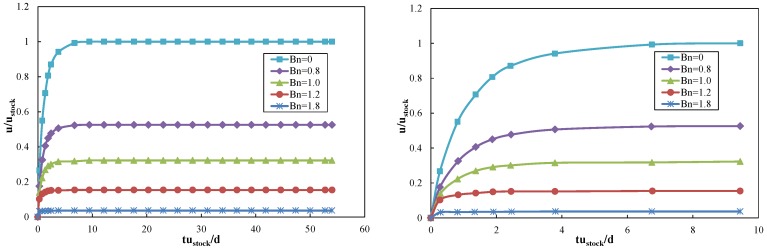
Falling velocities of particle in Bingham fluid changing with time (left: total process; right: in the beginning).

**Figure 5 materials-12-03128-f005:**
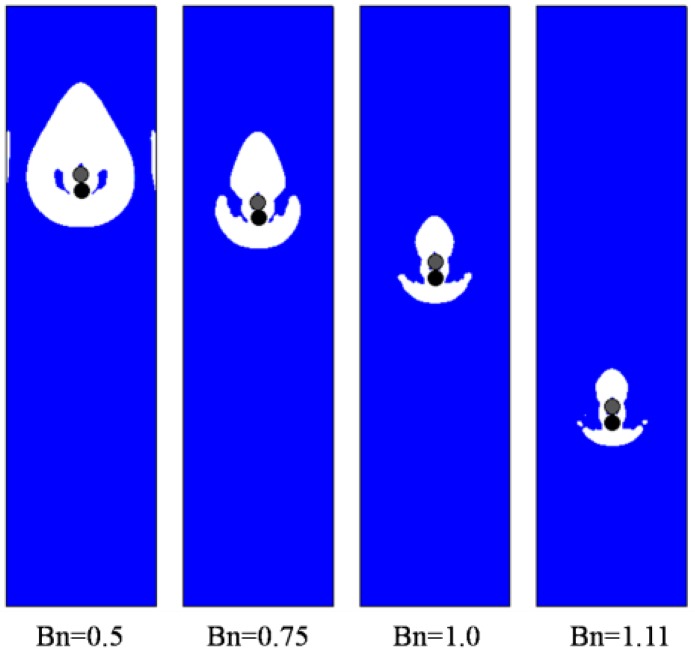
Snapshots at the kissing time of the two circular particles’ sedimentation processes in Bingham fluid with various *Bn*s (Yielded region: white; Unyielded region: blue).

**Figure 6 materials-12-03128-f006:**
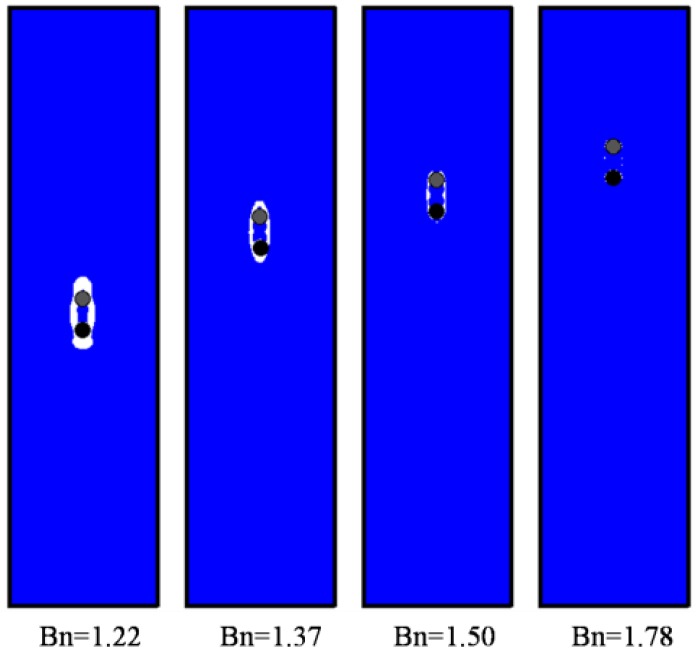
Snapshots at 20,000 time steps of two circular particles’ sedimentation in Bingham fluid with various *Bn*s (Yielded region: white; Unyielded region: blue).

**Figure 7 materials-12-03128-f007:**
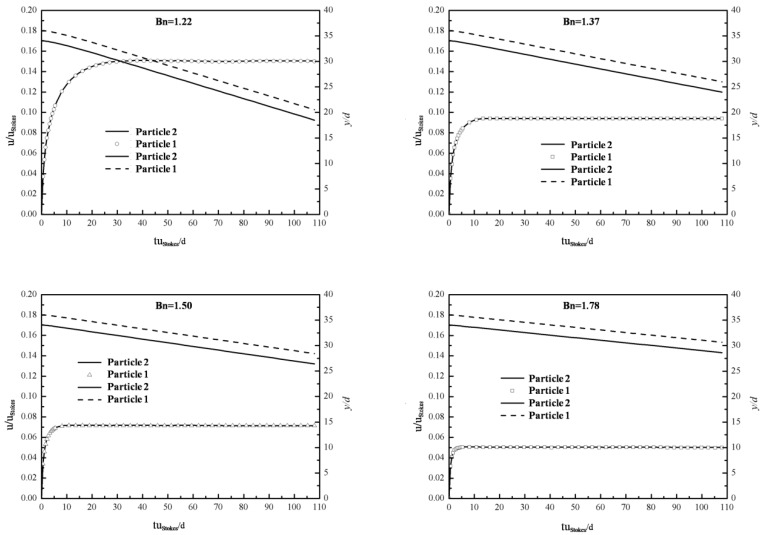
Velocities and positions of the two falling particles in Bingham fluid with various *Bn*s (1.22, 1.37, 1.50, and 1.78).

**Figure 8 materials-12-03128-f008:**
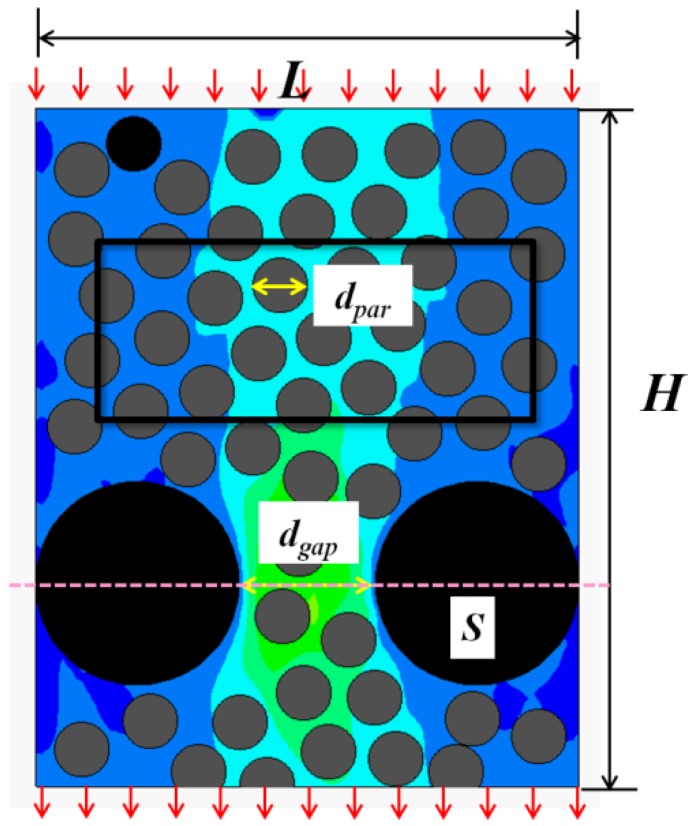
Simulation layout of SCC flow in a single channel.

**Figure 9 materials-12-03128-f009:**
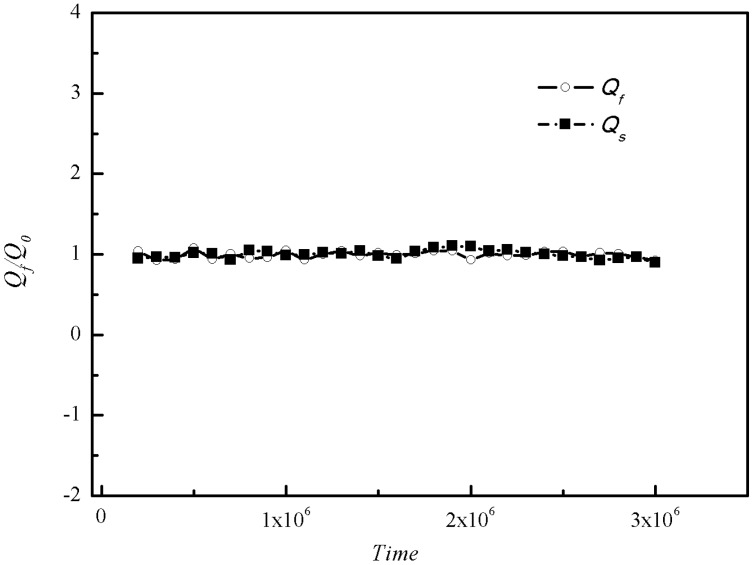
The statistics results of fluxes of fluid *Q_s_* and particles *Q_f_* when *Ar* = 8.

**Figure 10 materials-12-03128-f010:**
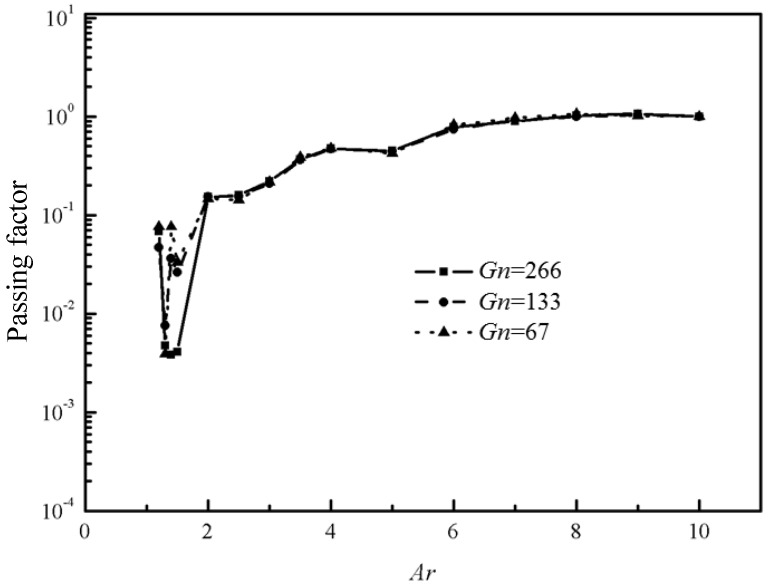
The passing factor *PF* with different *Gn* (*ϕ_s_* = 0.46).

**Figure 11 materials-12-03128-f011:**
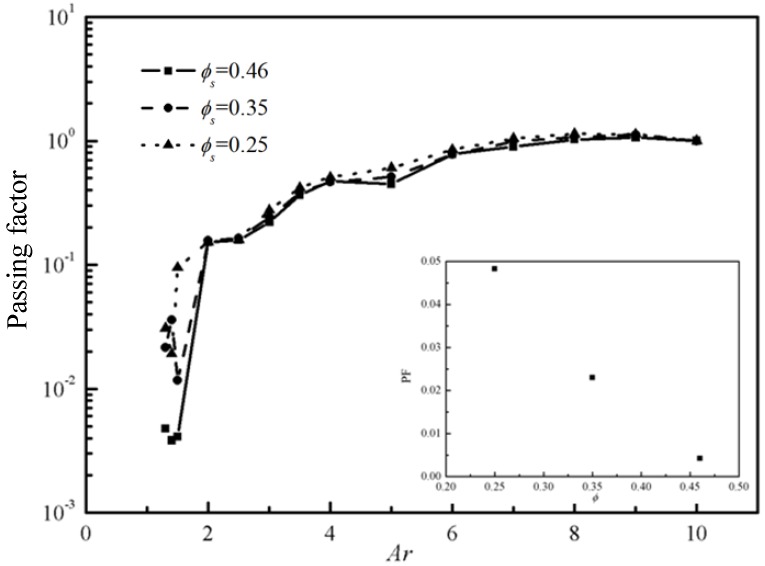
The passing factor *PF* with different *ϕ_s_* (*Gn* = 266).

**Figure 12 materials-12-03128-f012:**
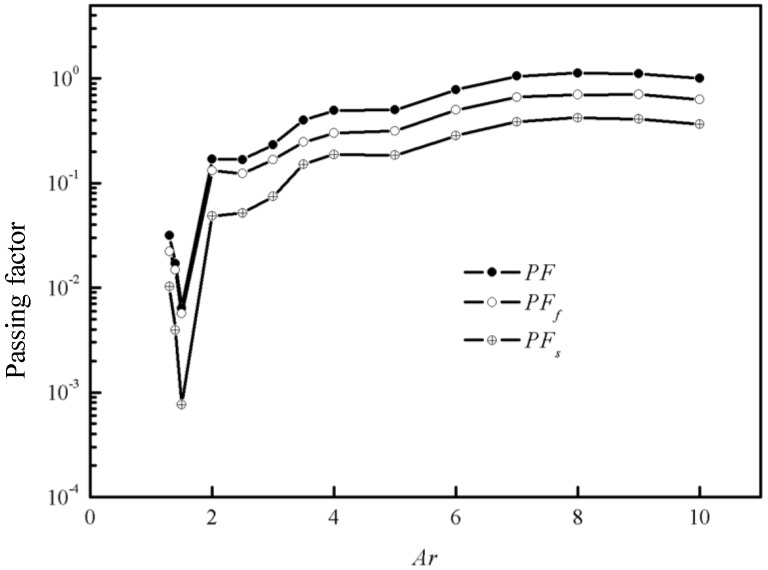
The passing factor *PF, PF_f_, PF_s_* when *Gn* = 133, *ϕ_s_* = 0.35.

**Figure 13 materials-12-03128-f013:**
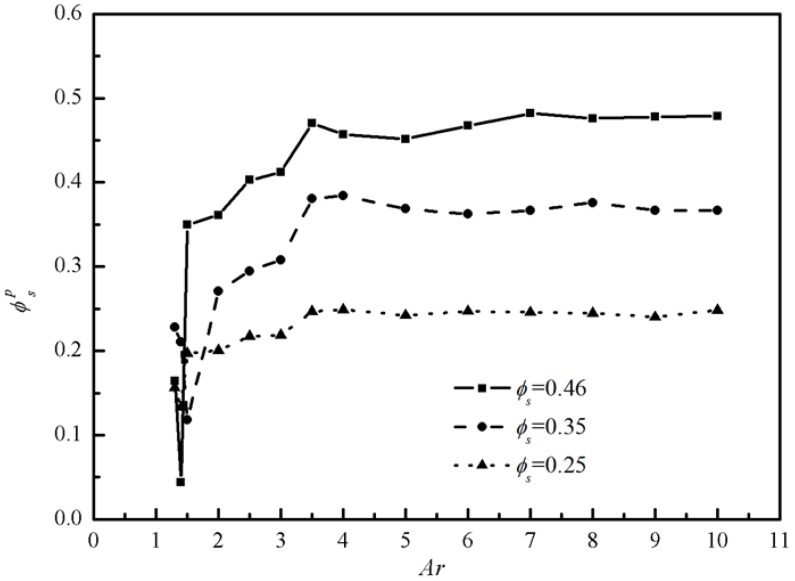
The solid volume fraction of the two-phase flow behind the cross section *S* with different *ϕ_s_*.

**Figure 14 materials-12-03128-f014:**
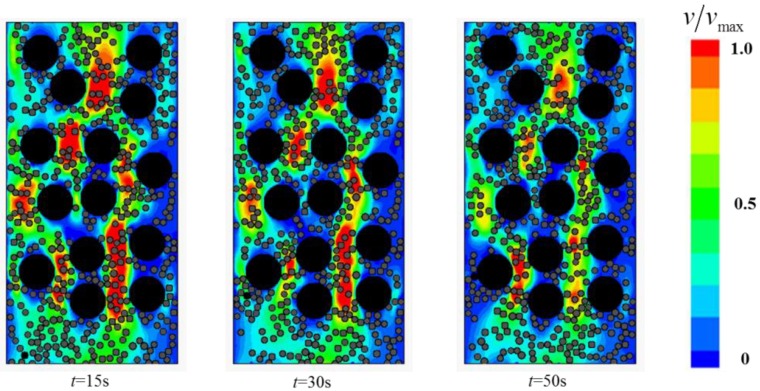
The flow field at different times with 4 narrow channels.

**Figure 15 materials-12-03128-f015:**
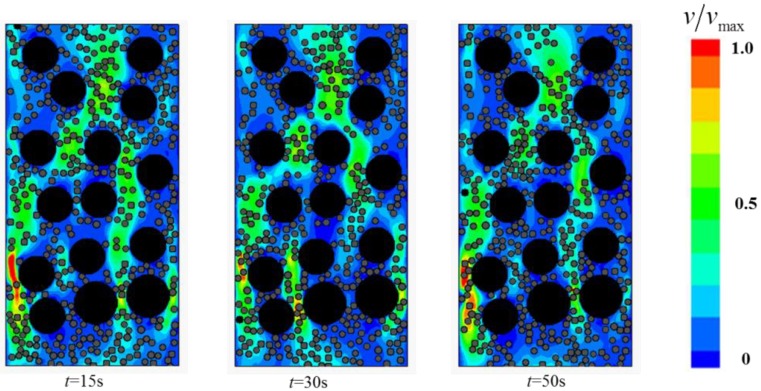
The flow field at different times with 8 narrow channels.

**Figure 16 materials-12-03128-f016:**
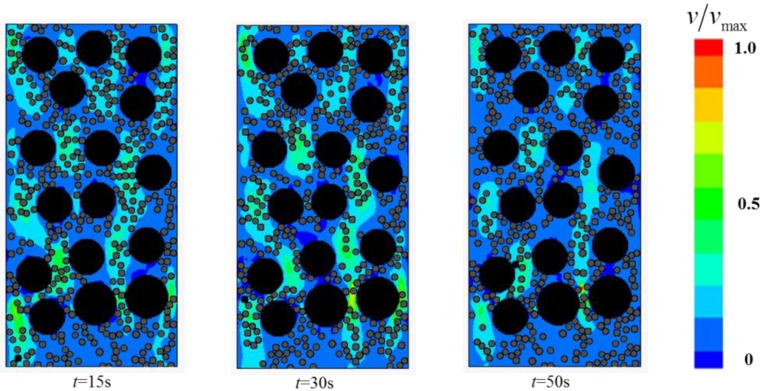
The flow field at different times with 10 narrow channels.

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
