# Peer review of "Lattice Boltzmann-Discrete Element Modeling Simulation of SCC Flowing Process for Rock-Filled Concrete"

_materials, 2019, doi:10.3390/ma12193128_

Round 1
Reviewer 1 Report
This paper investigated the passing ability of SCC based on coupling application of LBM method and DEM method. This is an interesting research. Kindly addressing the following drawbacks could be helpful for improving the manuscript’s quality:
1. The introduction should identify the gap the paper is trying to fill and why it is important; a background section should describe performed testing, main outcome, and physical meaning of the outcome; and the methodology should detail the tested materials and the approach in the study.
2. There are few references can be found in introduction part. Besides, more Quantitative indicators from previous studies are supposed to be supplemented in this part for aggregate shape discussion. It is obvious that there are many practical and accurate ways to do the simulation of pavement materials, this section cannot be found in background section, the paper listed follows can be taken as one instance [1,2]. The shape description of the morphological indexes from aggregates should also take into account in DEM simulation. The aims have been frequently used for particles shape characterization. [1] Wang H, Wang C, You Z, Yang X, Huang Z. Characterising the asphalt concrete fracture performance from X-ray CT Imaging and finite element modelling. International Journal of Pavement Engineering. 2018 [2] Wu W, Tu Z, Zhu Z, Zhang Z, Lin Y. Effect of gradation segregation on mechanical properties of an asphalt mixture. Applied Sciences, 2019
3. The results and conclusion section should go beyond summarizing the test outcome and explain the reason behind the results, and the understanding gained from those results. 4. For Section 3 Validation of the numerical model, Please provide more details on the referred research (Feng Z.-G.& Michaelides 151 E.E.(2005).
Author Response
Please kindly find the responses in the attached file.

Reviewer 2 Report
This paper is presenting Lattice Boltzmann-discrete element modelling simulation of SCC flowing process for rock-filled concrete. More analytical and experimental backgrounds of rock-filled concrete and why modelling it with Lattice Boltzmann-discrete element is important should be presented. The current format of the paper is not acceptable.
Author Response
Please kindly find the attached responses.

Reviewer 3 Report
This work focuses on the simulation of self-compacting concrete (SCC) flows in the porous rock-filled concrete using coupled lattice Boltzmann and discrete element method. The authors first validate the model and applied it to study SCC in a single channel and a porous medium.
The manuscript presents an interesting topic. However, the authors have to make sure its text is grammatically correct. The manuscript can be improved in my opinion, as detailed bellow. With appropriate consideration to the suggested comments and grammatical revision as well as other reviewers' comments, I find this paper appropriate for publication in Materials journal.
Please modify Eq. 1. In lines 166-167 please clarify which particle is the first particle and which one is the second particle? Provide a reference for the “Drafting, Kissing and Tumbling (DKT) process” that you mentioned in line 170. Some other studies on implementing LBM for fluid flow simulation may also be cited, such as
“Computer simulation of the effect of deformation on the morphology and flow properties of porous media”
“Pore-scale characteristics of multiphase flow in heterogeneous porous media using the lattice Boltzmann method”
In section 3.1, it seems that you compared your results with Feng et al. (2007), as mentioned in line 173. However, you refer to it as Feng et al. (2004) in Figs. 2 and 3. Please clarify. I would recommend explaining which part of the Figs. 2 and 3 correspond to the kissing process, and which part represents the tumbling process. Please explain Feng’s method. Is it based on a numerical simulation method or an analytical approach? If it is based on a simulation approach, I recommend the authors to validate their method by using an analytical model. Markers in Fig. 4 are not clear enough. You can use different color and larger markers to help the reader catch the Fig. easily. Explain what the yielded region means (line 239). I wonder if you have compared your results in section 3.2 with any theoretical model. Since it is introduced as model validation, it needs to be validate with a model. Show a sample of “main stream” and “tributary stream” in Figs. 14-16.

Author Response
Please kindly find the file attached.

Round 2
Reviewer 1 Report
Now, this manuscript is good for publication.
Author Response
The authors appreciate the positive attitude from the reviewer on our manuscript.
Reviewer 2 Report
The paper has improved, however I think the literature review can be extended and has more related studies as below:
Effects of specimen size and shape on compressive and tensile strengths of self-compacting concrete with or without fibers. Magazine of Concrete Research, 65(15), 914-929.
Nanoparticles in self-compacting concrete - a review. Magazine of Concrete Research, 67(20), 1084-1100.
Author Response
The comments have been followed accordingly.
Reviewer 3 Report
The authors addressed the comments completely.
Just one point: Eq. 1 needs revision.